# Progesterone: A Neuroprotective Steroid of the Intestine

**DOI:** 10.3390/cells12081206

**Published:** 2023-04-21

**Authors:** Lennart Norman Stegemann, Paula Maria Neufeld, Ines Hecking, Matthias Vorgerd, Veronika Matschke, Sarah Stahlke, Carsten Theiss

**Affiliations:** 1Department of Cytology, Institute of Anatomy, Ruhr-University Bochum, D-44801 Bochum, Germany; lennart.stegemann@rub.de (L.N.S.); paula.neufeld@rub.de (P.M.N.); ines.hecking@rub.de (I.H.); veronika.matschke@rub.de (V.M.); sarah.stahlke@rub.de (S.S.); 2Department of Neurology, Neuromuscular Center Ruhrgebiet, University Hospital Bergmannsheil, D-44789 Bochum, Germany; matthias.vorgerd@bergmannsheil.de

**Keywords:** progesterone, ENS, PR-A/B, PGRMC1, mPRa, mPRb, AG205, neurosteroid, rotenone, Parkinson’s disease, LMD

## Abstract

The enteric nervous system (ENS) is an intrinsic network of neuronal ganglia in the intestinal tube with about 100 million neurons located in the myenteric plexus and submucosal plexus. These neurons being affected in neurodegenerative diseases, such as Parkinson’s disease, before pathological changes in the central nervous system (CNS) become detectable is currently a subject of discussion. Understanding how to protect these neurons is, therefore, particularly important. Since it has already been shown that the neurosteroid progesterone mediates neuroprotective effects in the CNS and PNS, it is now equally important to see whether progesterone has similar effects in the ENS. For this purpose, the RT-qPCR analyses of laser microdissected ENS neurons were performed, showing for the first time the expression of the different progesterone receptors (PR-A/B; mPRa, mPRb, PGRMC1) in rats at different developmental stages. This was also confirmed in ENS ganglia using immunofluorescence techniques and confocal laser scanning microscopy. To analyze the potential neuroprotective effects of progesterone in the ENS, we stressed dissociated ENS cells with rotenone to induce damage typical of Parkinson’s disease. The potential neuroprotective effects of progesterone were then analyzed in this system. Treatment of cultured ENS neurons with progesterone reduced cell death by 45%, underscoring the tremendous neuroprotective potential of progesterone in the ENS. The additional administration of the PGRMC1 antagonist AG205 abolished the observed effect, indicating the crucial role of PGRMC1 with regard to the neuroprotective effect of progesterone.

## 1. Introduction

Progesterone, commonly known as a sexual hormone, is a steroid hormone and one of the most important progestogens [1]. In addition to its major role in the regulation of female reproduction, it is believed to have effects in cancer biology [2] and is relevant as a neurosteroid in the central (CNS) [3] and peripheral nervous system (PNS) [4,5]. In both the CNS and PNS, progesterone shows diverse neuroprotective and neuroplastic effects [1,6], while a very recent study shows for the first time that progesterone also has a neuroprotective effect in the enteric nervous system (ENS) [7]. The ENS is an autonomous network of about 100 million neurons embedded in the wall of the gut, organized into two main plexus named submucosal and myenteric plexus. A wide variety of neurotransmitters, functionally distinct enteric neurons and glia cells closely resemble the complexity of the CNS, which is the reason why it is often referred to as the “second brain” [8,9]. Considering recent studies about the pathogenesis of neurodegenerative disorders, such as Parkinson’s disease (PD), the neuroprotective potential of progesterone in the ENS becomes of particular interest. Specifically, the gut–brain axis, which is a bidirectional communication network between the ENS and CNS [10,11], seems to be an important factor for the pathogenesis. For example, there is evidence that misfolded alpha-synuclein aggregates, known as Lewy bodies, prime in the ENS and spread to the CNS via the vagal nerve, where it may lead to PD [12,13,14]. The cause for the typical hypokinetic PD symptoms is a loss of dopaminergic neurons from the brainstem area of the CNS. This characteristic loss of dopaminergic neurons in the CNS in PD has also been demonstrated in the human ENS [15]. Right now, four different toxin-induced models are used to simulate PD, such as 6-hydroxydopamine (6-OHDA), 1-methyl-4-phenyl-1,2,3,6-tetrahydropyridine (MPTP), paraquat and rotenone. They all generate oxidative stress, which leads to cell death in dopaminergic (DA) neuronal population in vivo [16], and therefore, induces PD symptoms on a cellular and behavioral level. Rotenone is observed to additionally induce alpha-synuclein aggregates [17,18]. Despite these recent developments, there has been insufficient focus on research on progesterone and its neuroprotective potential in the ENS. However, the extent to which progesterone may also have a neuroprotective effect in the ENS in the context of neurodegenerative diseases has not yet been sufficiently analyzed. For this reason, the present study investigates for the first time whether and which progesterone receptors are expressed in the ENS at different time points during development. Subsequently, the neuroprotective potential of progesterone on rotenone-treated neurons were investigated in dissociated ENS cultures and potential mechanisms of action were discussed.

## 2. Materials and Methods

All experiments in this study have been performed in strict accordance with institutional, German (TschG) and EU guidelines for the care and use of laboratory animals.

### 2.1. Preparation for Cryosectioning

Intestines of Wistar rats at the postnatal (p) day 9, 15 and 30 were used for the methods previously described by Hecking et al. [19]. All solutions were prepared with DEPC-treated water (#D5758-50ML, Sigma Aldrich, St. Louis, MO, USA). Preparation tools and Petri dishes were sterilized for 4 h at 240 °C, and the work surfaces were cleaned with NaOH-EDTA (0.1 M NaOH, 1 mM EDTA). The small intestine was extracted using the techniques shown in videos in Hecking et al. [19]. Several 2–3 cm long pieces of the intestines were placed on microscope slides (#AAAA000001##12E, Thermo Scientific, Waltham, MA, USA) covered with aluminum foil and shock-frozen at −50 °C with the aid of the LIENS chamber. The frozen pieces were then transferred onto the specimen holder, cut into 10 μm thick sections in the cryostat (CryoStar NX50, Thermo Scientific) and mounted on the required slide.

### 2.2. Laser Microdissection and RT-qPCR

The gene expression of the different progesterone receptors in the myenteric plexus at different developmental stages (p9, p15 and p30) were analyzed using laser microdissection (LMD) and subsequent RT-qPCR. Before isolating the myenteric ganglia via LMD, the slides (#11505151, Leica Microsystems, Wetzlar, Germany) were stained with a RNase-free 0.5% cresyl violet solution (#7651.1, Carl Roth, Karlsruhe, Germany). The system settings of the Leica LMD6500 proposed in Hecking et al. [19] were used for cutting out the ganglia. In brief: power 34; aperture 17; speed 33; specimen balance 18; 20× magnification. For each RT-qPCR, approximately 7 mm^2^ of isolated myenteric ganglia were needed to extract RNA concentrations of about 900 ng/μL using the Monarch^®^ Total RNA Miniprep Kit (#T2010S, New England BioLabs, Ipswich, MA, USA) and achieving cDNA concentrations of about 400 ng/μL with the GoScriptTM Reverse Transcription Mix, Oligo(dT) (#A2790, Promega, Madison, WI, USA). Both kits were used according to the manufacturer’s protocol. In the following RT-qPCRs, the expression levels for the housekeeping genes *GAPDH*, *NR3C3* (PR-A/B), *PAQR7* (mPRa), *PAQR8* (mPRb) and *PGRMC1* (see Table 1) were measured in triplicate and in at least three independent runs. As in Hecking et al. [19], 10 μL GoTaq qPCR Master Mix (#A6001, Promega), 1.4 μL primer upstream, 1.4 μL primer downstream, diluted cDNA and ddH2O were combined to a final volume of 20 μL. Using the CFX96 Real Time PCR Detection System (BioRad, Hercules, CA, USA) samples were heated to 95 °C for 2 min, followed by 40 amplification cycles, 15 s at 95 °C and 60 s at 60 °C.

### 2.3. Immunohistochemistry

Immunofluorescence staining and confocal laser scanning microscopy (Zeiss, Jena, Germany, LSM 800) at 40× of the myenteric ganglia was performed using the protocol described in Hecking et al. [19]. The cryosections of the intestines were collected on Superfrost-Plus Adhesion Slides (#J1800AMNZ, Thermo Scientific) and fixed with 4% PFA in phosphate-buffered saline (PBS) for 15 min. After rinsing with PBS for 3 × 5 min, the tissue was permeabilized with 1% Triton-X-100 (#T8532; Sigma-Aldrich, St. Louis, MO, USA) in PBS for 15 min, followed by 3 × 5 min washing steps. The non-specific binding sites were blocked with goat serum (#G9023, Sigma-Aldrich; 1:50 in PBS) for 30 min, which was removed with a 2 min washing step with PBS. Next, the cryosections were incubated overnight with the primary antibody at 4 °C. For the visualization of myenteric neurons, beta-III-tubulin (TUJ-1, #MAB1195, RD-Systems; 1:500 in PBS) or PGP 9.5 (#PA1-10011, Thermo Fisher, 1:200 in PBS with 5% goat serum) were used as specific antibodies. After two further washing steps with PBS (2 × 10 min), secondary antibodies, here, goat anti-mouse (#T5393, Sigma-Aldrich; 1:750 in PBS) goat anti-chicken IgY (#A11041, Invitrogen, Waltham, MA, USA; 1:400 in PBS), were applied for 2 h and then removed with 2 × 10 min PBS washing cycles. A second primary antibody (#MA1-410 (PR-A/B), #A561376 (mPR), #A5120929 (PGRMC1), Thermofisher; 1:400 in PBS) to mark progesterone receptors was applied over night at 4 °C and removed as described above. This was followed by an incubation with the appropriate second secondary antibody (e.g., #A11001, Thermofish; 1:750 in PBS) for 2 h and the sequential washing step. In the last step, counterstaining was performed with DAPI for 30 min (#D9542. Sigma-Aldrich; 1: 5000 in PBS). Before the samples were finally embedded with a fluorescence mounting medium (#S3023, Dako, Glostrup, Denmark), intensive washing steps with PBS (3 × 5 min, 2 × 10 min) were performed.

### 2.4. Cell Culture

The myenteric plexus cells of male Wistar rats (p15) were cultivated after several steps of preparation and enzymatic digestion as previously described in Hecking et al. [19]. The preparation tools and Petri dishes were sterilized at 240 °C for 4 h, and the work surfaces were cleaned with NaOH-EDTA (0.1 M NaOH, 1 mM EDTA) beforehand. All solutions were prepared under sterile conditions at an appropriate workbench to avoid contamination. They were sterilized by membrane filtration (0.2 μm) or autoclaving. A detailed description of each solution can be found in Appendix A gives an overview of the cultivation process.

### 2.5. Preparation and Cultivation

Prior to the preparation, the coverslips were coated with poly-d-lysine (#P7280-5MG, Sigma-Aldrich) overnight at 4 °C and washed once with PBS. Three male Wistar rats (p15) were used per preparation. First, the intestines were removed and stretched onto a Combitip Plus tip (#p1877, Eppendorf, Hamburg, Germany), as shown in Hecking et al. [19], which was used to ease the detachment of the tunica muscularis with forceps. The isolated muscularis was kept in Hanks’ Balanced Salt Solution (HBSS, #HS264-500ML, Sigma-Aldrich) until the preparation was completed. After the careful removal of the liquid, the myenteric cells were enzymatically digested with 0.05% trypsin + EDTA (#25300054, Gibco). A total of 5 mL of trypsin 0.05% + EDTA was added to the tissue and then mixed with a heated magnetic stirrer at 400 rpm for five minutes. The supernatant was collected in a 50 mL Sarstedt Falcon (#62.547.254, Sarstedt, Nuembrecht, Germany) containing 20 mL of neurobasal medium (#10888022, Thermo Fisher) with 10% fetal bovine serum (#F7524, Sigma-Aldrich) and 0.01% penicillin–streptomycin (#P4333, Sigma-Aldrich). A total of 5 mL of trypsin 0.05% + EDTA was added to the remaining tissue, the procedure was repeated and the supernatant was added to the 50 mL Sarstedt Falcon. In total, this step was repeated four times until 20 mL of supernatant was collected. The cells were then separated from the medium using a centrifuge (5 min, 1000 rpm, RT). After removing the supernatant, the cells were resuspended in 12 mL proliferation medium and spread onto a 24-well plate (#83.3922, Sarstedt). Seeding was followed by a total incubation period of 10 days at 37 °C and 5% CO_2_, with medium changed every 3 days (Figure 1). On Day 6, the proliferation medium was replaced with a differentiation medium. On Day 9, external factors were administered.

### 2.6. Intoxication with Rotenone

On Day 9, the differentiation medium was replaced by a medium containing the external factors. The wells were divided into five groups. The groups were as follows: control group, rotenone 1 nM (#45656, Sigma-Aldrich), progesterone 10 nM (#P8783. Sigma-Aldrich), rotenone 1 nM + progesterone 10 nM, rotenone 1 nM + progesterone 10 nM + AG205 5 nM (#A1487, Sigma-Aldrich) and AG205 5 nM only. All wells contained a final concentration of 1 µL/mL propidium iodide (PI) (#P4864, Sigma-Aldrich). Rotenone and AG205 were diluted in DMSO until the desired concentration was reached. Progesterone was diluted in EtOH. Note that the final concentration of DMSO and EtOH in the medium was never higher than 0.01% respective 1%, which has no influence on cell integrity [20]. All external factors were stored at −20 °C. The cell culture was then incubated for 24 h and fixed with 4% PFA in PBS for 15 min under dark conditions, as PI is light sensitive.

### 2.7. Immunohistochemistry, Counting Method and Statistics

After rinsing the slides with PBS, the cell membranes were permeabilized with 0.3% Triton-X-100 (#T8532, Sigma-Aldrich) in PBS for 15 min. After washing the coverslips with PBS (3 × 5 min), non-specific binding sites were blocked by incubation with goat serum (#G9023, Sigma-Aldrich; 1:50 in PBS) for 30 min. After washing (2 × 5 min), the cells were incubated with a primary antibody against beta-III-tubulin (TUJ-1, #MAB1195, RD-Systems, Singapore; 1:500 in PBS) at 4 °C overnight. After extensive washing (3 × 5 min) with PBS, a matching secondary antibody (goat anti-mouse, #A11001, Molecular Probes; 1:1000 in PBS), here, in green, was applied for 2 h, followed by further washing steps (3 × 5 min). The nuclei were then stained with DAPI (#D9542, Sigma-Aldrich; 1:1000 in PBS) for 30 min. Additional washing steps with PBS (3 × 5 min) were performed before embedding the coverslips in fluorescence embedding medium (#S3023, Dako). Images were acquired with a Keyence fluorescence microscope (BZ-X800) at a magnification of 60×, using the same exposure time in the different channels for each coverslip (DAPI: 1/4 s, tuj1: 1/2.5 s, PI: 1/2 s). The imaging and counting of the different cell types were performed in a strictly blinded manner to avoid biased results. Cells stained with PI were counted as dead. Since the data had a Gaussian distribution and an equal standard deviation, the survival rate of cells in the different experimental groups was compared using a one-way ANOVA and a Tukey’s multiple comparison test. Microsoft Excel (Version 16.72, Microsoft Corporation, Redmond, WA, USA) and GraphPad Prism version 9 (GraphPad, San Diego, CA, USA) were used for statistical analysis.

## 3. Results

### 3.1. Progesterone Receptor mRNA Expressions in the Myenteric Plexus of Rats

The RT-qPCR analysis confirmed the expression of PR-A/B, mPRa, mPRb, PGRMC1 mRNA in laser the microdissected myenteric ganglia of rats at different developmental stages (p9, p15 and p30). Figure 2a displays the different expression levels normalized to the housekeeping gene GAPDH. The PGRMC1 gene showed the highest relative expression level (mean ΔCt^−1^ = 0.168) along all ages, followed by mPRa (0.103) and mPRb (0.076). PR-A/B showed the lowest relative mRNA expression at all times investigated (0.067). Figure 2b indicates that the receptors are constantly expressed during development, without significant change of expression levels.

### 3.2. Progesterone Receptors Are Expressed on Protein Level

We could confirm the expression of PR-A/B (Figure 3a), mPRs (b) and PGRMC1 (c) on protein level by immunostaining and confocal laser scanning microscopy in rat small intestine. The colocalizations of the receptors and myenteric neurons are shown in Figure 3 using p30 as an example, as there are no differences at the different ages in line with the results of the mRNA expression. Moreover, a low signal of PR-A/B expression in the surrounding muscularis was detected, which is in accordance with a previous study [21].

### 3.3. Reduction in Cell Death In Vitro

A possible neuroprotective effect of progesterone in ENS neurons treated with rotenone was analysed by counting dead neurons labelled with PI in relation to the total number of neurons. The following data are normalised to untreated controls. The survival of ENS neurons in the rotenone-treated group (1 nM rotenone, 24 h incubation) decreased by 70% (mean 0.26 ± 0.22 compared to controls with mean 0.96 ± 0.31). In cultures treated with rotenone plus progesterone, the percentage of surviving neurons decreased by only 25% compared to controls (0.71 ± 0.41) (Figure 4i). Thus, treatment with 10 nM progesterone resulted in a significant increase in cell survival of over 45% compared to rotenone-treated ENS neurons without additional progesterone treatment (*p* < 0.0001). Incubation of these neurons with a combination of the rotenone + progesterone group plus the PGRCM1 antagonist AG205 again reduced survival to 22.97% (0.23 ± 0.22). This represents a significant reduction in ENS neuron survival compared to the rotenone + progesterone group (without PGRMC1 antagonists) (*p* < 0.0001). Administration of progesterone or AG205 alone had no significant effect on ENS cell survival compared to untreated controls (mean progesterone 1.09 ± 0.49, mean AG205 0.81 ± 0.53).

## 4. Discussion

Although the ENS has generally received increased public and scientific attention in recent years, concrete studies on the potentially neuroprotective effects of hormones such as progesterone on ENS neurons have been investigated in only one paper to date [7]. In this important work, progesterone showed neuroprotective and anti-inflammatory effects in the ENS in a Parkinson’s disease mouse model. The present work now complements and extends these data by systematically investigating progesterone receptor expression in the ENS over different juvenile developmental time points and analyzes neuroprotection in an ENS cell culture model in which cell stress is induced using rotenone.

Here, we show for the first time that the progesterone receptors PR-A/B, mPRa, mPRb and PGRMC1 are consistently expressed at the mRNA level in the rat myenteric plexus during development (p9, p15 and p30). The mRNA expression levels of each receptor in the ENS do not change significantly during aging. Interestingly, PR-A/B showed the lowest mRNA expression levels. In addition, we detected higher relative expression levels of mPRa mRNA compared with mPRb. PGRMC1 showed the highest expression at the mRNA level across all ages tested. At the protein level, we detected the colocalization of these progesterone receptors in myenteric ganglia at each developmental stage using confocal laser scanning microscopy. Because there are currently no other data for the ENS available, we also compared these results with data available for the CNS. The classical progesterone receptors PR-A/B can be detected in different neuronal cell types of the rat brain [22,23,24], but unlike in the ENS, their regional expression varies as does their expression during aging [25]. In most cerebral regions, PR-A/B appears to be somewhat more highly expressed at older developmental stages [26], whereas PR-A/B is more highly expressed in the cerebellum of newborn rats [27]. This age-dependent expression being caused by epigenetic regulations such as post-transcriptional interventions [28,29,30] is a subject of discussion.

It can currently only be speculated why there is no age-dependent regulation of progesterone receptor expression in the ENS at least during the first 30 days. It is possible that progesterone is not as important to the developmental processes of the ENS neurons as it is in the CNS. Concerning the comparison of the mRNA levels of the individual receptors in contrast to the ENS, mPRb is more pronounced than mPRa in the CNS [31]. However, just like in the ENS, the mRNA levels of both membrane receptors are lower than the expression levels of the PGRMC1 mRNA in the CNS [32]. This comparison of relative receptor expression at the mRNA level might lead to the assumption that PGRMC1 is the predominant receptor mediating the rapid neuroendocrine effects in the ENS, as it is also believed to be in most cerebral regions [32] and in the spinal cord [33]. Whether these receptors also show different levels of protein expression needs to be further investigated. It has also not yet been clarified whether there are differences between male and female rats regarding receptor expression and to what extent the expression of progesterone receptors in enteric neurons is independent of progesterone treatment, as it has already been shown in the rat brain [34]. Especially for such studies, there should be sex-specific examinations of adult rats (older than 60 days).

In the present study, we were able to significantly reduce the rotenone-induced cell death of cultured myenteric neurons by 45%. Complex I inhibitor rotenone is an established model to reproduce features of PD in a dose-dependent manner in vivo and in vitro. By blocking the mitochondrial NADH-dehydrogenase, it leads to ATP depletion, oxidative stress and, ultimately, cell death [17]. Additionally, rotenone causes the aggregation of alpha-synuclein and, when administered peripherally, a peripher-to-central transmission of the aggregated protein has been shown [35,36]. The dose of 10 nM progesterone used is in agreement with Peluso et al., who have shown that this is the maximum effective dose for inhibiting apoptosis in spontaneously immortalized granulosa cells (SIGCs) [37]. Since only male animals were used to obtain the present data on the neuroprotective effect of progesterone in ENS cell culture treated with rotenone, further studies are needed to evaluate the role of sex differences. Sex is a known mediator of the onset and progression of neurodegenerative diseases, as well as the consequences of neurologically relevant events, such as stroke or traumatic brain injury [38,39,40]. As progesterone levels naturally differ between the biological sexes, further research is needed to investigate the influence of sex on progesterone and its binding sites in the context of the CNS and PNS.

Our results are in line with those of Jarras et al. from 2020 [7], where progesterone administration also prevented the loss of dopaminergic enteric neurons in MPTP-treated mice. Since the additional administration of the PCRMC1 antagonist AG205 abolished the neuroprotective effect in the current study, it is suggested that the neuroprotective effect of progesterone is not genomic in nature and can be attributed, at least in large part, to the interaction of progesterone with PGRMC1. The functional analyses of PGRMC1 mutations induced experimentally by small interfering ribonucleic acid (siRNA) treatment provides genetic evidence for the crucial role of PGRMC1 in binding progesterone and conferring its anti-apoptotic effect [41]. Progesterone is not thought to bind directly to PGRMC1 alone. Instead, the plasminogen activator inhibitor 1 RNA-binding protein (PAIRBP1), also known as serpin mRNA-binding protein (SERBP1), and PGRMC1 form a complex that acts as a membrane receptor for progesterone [42]. Recent studies suggest the involvement of PAIRBP1 as a downstream component of the pathway rather than the formation of a complex as a component of the membrane receptor [43]. However, while the exact mechanism remains to be further explored, there is strong evidence for an interaction of PAIRBP1 and PGRMC1 in mediating the neuroprotective effects of progesterone [44]. While PAIRBP1 has been shown to be expressed in different regions of the rat brain [32], the expression of PAIRBP1 in the cells of the enteric nervous system has not yet been studied and remains a question to be addressed in the future.

Genomic signaling cascades as mediated by the classical receptor PR-A/B, which mediate their effect as hormone-dependent transcription factors [45,46,47], are probably not mainly responsible for neuroprotection in the rotenone model, as their effects are usually mediated after more than 24 h. The non-genomic PR-A/B-induced inhibition of the phosphoinositide-3-kinase/protein kinase B (PI3K/Akt) signaling pathway was already detected after 24 h in the rat brain. This signaling pathway regulates inflammation and cell survival [48,49], which could be important in countering the rotenone-induced activation of NLRP3 inflammasomes and pyroptosis [50]. Further non-genomic mediated neuroprotective effects of PR-A/B could be explained due to the mitogen-activated protein kinase (MAPK) pathway previously demonstrated in the rat brain [51]. However, the observed attenuation of the neuroprotective effect of progesterone after the administration of the PGRMC-1 antagonist AG205 rather suggests a strong involvement of PGRMC1.

PGRMC1 modulates cytochrome P450 enzymes and seems to be involved in the regulation of neurosteroidogenesis in the CNS [5]. Furthermore, PGRMC1 is believed to regulate intracellular calcium levels and to increase BDNF levels via the modulation of ERK-signaling [52]. Due to the fact that PGRMC1 is associated with the membranes of the Golgi apparatus, the ER and mitochondria, this receptor could be important for the protection against stress in endoplasmic reticulum and mitochondria triggered by rotenone [18]. In particular, the inhibition of complex I by rotenone can lead to the opening of the mitochondrial permeability transition pore (PTP) and massive production of reactive oxygen species (ROS), which play an important role in rotenone toxicity [53,54]. The decrease in cell death after progesterone treatment may, therefore, also be explained by the anti-oxidative effects of progesterone probably mediated by PGRMC1 [55].

Studies correlating the structural components of PGRMC1 with its function found that the D120G mutation leads to loss of progesterone-dependent anti-apoptotic properties [41]. These data are consistent with clinical observations that a D120G mutation of PGRMC1 in breast cancer cells results in a more sensitive response to the apoptotic effects of chemotherapeutic agents [56,57].

The PGRMC1 binding site, which most likely specifically binds progesterone, is located in the transmembrane domain and an adjacent segment of the C-terminus [41]. In silico analyses revealed that the C-terminus is most likely involved in signal transduction as it contains a heme-binding domain and several SH2 and SH3 segments [42].

In spontaneously immortalized granulosa cells, the activation of PGRMC1 by progesterone leads to the suppression of apoptosis-promoting genes and the increased expression of the gene encoding the anti-apoptotic BCL2AID, a member of the BCL2 family [37]. However, the exact molecular mechanisms involved in the neuroprotective effect of the interplay between PGRMC1 and progesterone in the ENS remain unclear and are, therefore, a major area for further investigation.

The exact roles of each receptor in the neuroprotective effects in this model appear to be very complex and cannot be fully elucidated with the current data. Nevertheless, our data strongly suggest that PGRMC1 plays a crucial role as a mediator of the neuroprotective effects of progesterone in the ENS. Further studies are required to improve the understanding of progesterone-mediated control circuits. Still, a similar neuroprotective potential of progesterone in the CNS was postulated based on experimental animal models [58]. These promising results could also be reproduced in two Phase II clinical studies, which examined progesterone administration after traumatic brain injury [59,60] but could not by confirmed in Phase III clinical studies of traumatic brain injury [61,62]. What seems to be an inconsistency may in fact be attributed to various factors. On the one hand, there are influencing factors regarding the injury, such as its heterogeneity, as well as possible comorbidities. On the other hand, factors regarding progesterone, such as dosing, administration and length of treatment, are found to be critical (reviewed in [63]). Taken together, a comprehensive understanding of the regulation and mode of action of the different progesterone receptors in the ENS is essential, first in the ENS cell culture model, then in the animal model and finally in humans.

## 5. Conclusions

The present work shows for the first time that the different progesterone receptors of the genomic and non-genomic signaling pathways are expressed in the neurons of the rat ENS. Here, PGRMC1 (non-genomic signaling pathway) displays the highest expression, which is not age-dependently regulated up to the age of 30 days. In ENS cultures treated with rotenone, progesterone was able to significantly reduce cell death. In ENS cell cultures, the antagonization of PGRMC1 by AG205 administration reverses the observed neuroprotective effect, supporting the conclusion that the neuroprotective effect of progesterone is mediated by non-genomic pathways involving PGRMC1. Even if it is not always easy to transfer research results from animals to humans, these data on the neuroprotective effect of progesterone in the ENS, which have only been generated in two different Parkinson’s models so far, can initially be considered promising.

## Figures and Tables

**Figure 1 cells-12-01206-f001:**
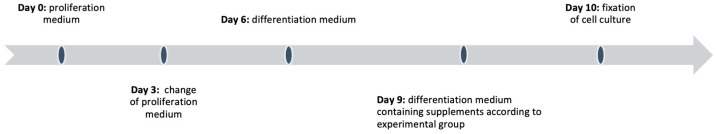
Overview of the different steps during cultivation of myenteric neurons.

**Figure 2 cells-12-01206-f002:**
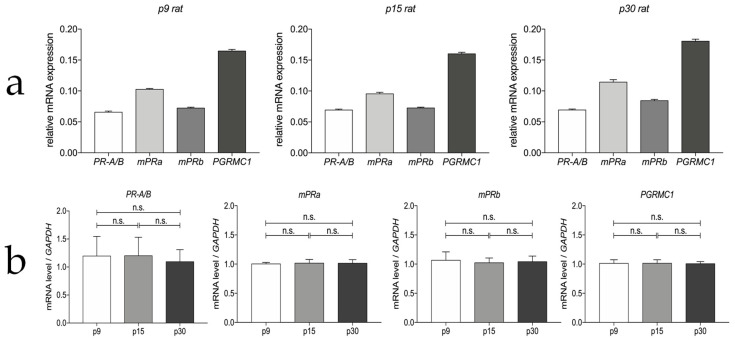
RT-qPCR analysis of progesterone receptor mRNA expression. (**a**) Comparison of relative concentration levels of each receptor mRNA at three different ages (p9, p15 and p30) using the reciprocal of ΔCt; (**b**) Comparison of the shift of expression levels of each receptor during development, which were normalized to the expression of the housekeeping gene GAPDH. Data are shown as mean ± SEM (n ≥ 3), with dissected myenteric ganglia of about 15 rats per age. ANOVA was used to compare the different ages, changes > 0.05 were considered significant, otherwise they were marked as not significant (n.s.).

**Figure 3 cells-12-01206-f003:**
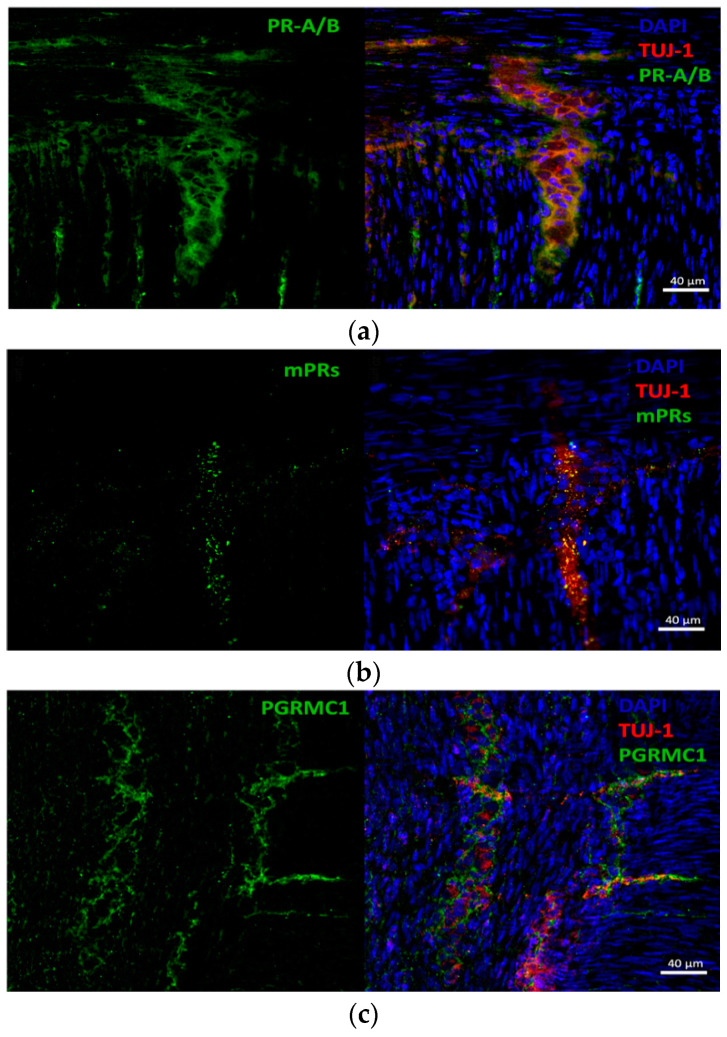
Confocal laser scanning microscopy myenteric ganglia in cryosections of Wistar rats at p30 for PR-A/B (**a**), mPR (**b**) and PGRMC1 (**c**). Cell nuclei were stained with DAPI in blue and the myenteric plexuses in red with TUJ-1. (**a**–**c**) All receptors (stained in green) are colocalized with the ENS neurons.

**Figure 4 cells-12-01206-f004:**
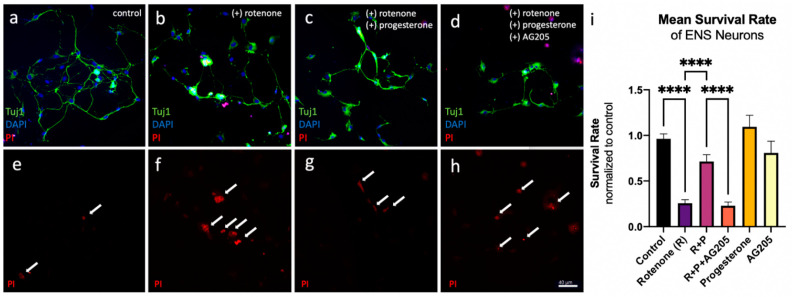
Confocal laser scanning microscopy of isolated plexus myenteric cells from male Wistar rats at p15 under different treatment conditions: (**a**,**e**) controls, (**b**,**f**) 1 nM rotenone, (**c**,**g**) 1 nM rotenone plus 10 nM progesterone, (**d**,**h**) 1 nM rotenone, 10 nM progesterone plus 5 nM AG205. (**a**–**d**) ENS neurons are marked in green (Tuj-1). (**e**–**h**) ENS neurons showing a positive PI signal (red) were counted as dead (white arrows). Scale bar: 40 µm. (**i**) Analysis of mean survival of ENS neurons treated for 24 h with rotenone (1 nM) with or without progesterone (10 nM) or AG205 (5 nM). Each experiment was performed in a blinded fashion. A total of 7415 cells were counted on 282 coverslips in 4 independent experiments. Statistical analysis was performed with GraphPad Prims software, using one-way ANOVA with a Tukey’s multiple comparison test, *p* < 0.0001 (****). Data are shown as mean ± SEM (*N* = 4).

**Table 1 cells-12-01206-t001:** Used primers for quantitative real-time PCR.

*GAPDH*	5′-ACT CCC ATT CTT CCA CCT TTG-3′, 3′-CCC TGT TGC TGT AGC CAT ATT-5′ (Microsynth)
*NR3C3* (PR-A/B)	5′-AGC ATG TCA GTG GAC AGA TG-3′, 3′-TAA GGC ACA GCG AGT AGA ATG-5′ (Microsynth)
*PAQR7* (mPRa)	5′-CCA CGG TTA TGC CTG AGA GT-3′, 3′-TAG TCC AGC GTC ACA GCT TC-5′ (Microsynth)
*PAQR8* (mPRb)	5′-AGA AGG GCT TCC CAA GAT GC-3′, 3′-AGT AGT AACGCC ACT CGT GC-5′ (Microsynth)
*PGRMC1*	5′-GAT GAC CTT TCT GAC CTCA CTC-3′, 3′-TTC CCA CGT GAT GGT ACT TG-5′ (Microsynth)

## Data Availability

Not applicable.

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
