# Peer review of "Progesterone: A Neuroprotective Steroid of the Intestine"

_cells, 2023, doi:10.3390/cells12081206_

Round 1

Reviewer 1 Report (Previous Reviewer 3)

Although PGRMC1 blockage experiment was added, the overall study is still too simple and lacks enough mechanistic explorations, especially in in vivo PD models. To strengthen the significance and novelty, further questions should be addressed, and additional experiments should be supplemented:

1. What is the expression of those progesterone receptors in ENS neurons of middle age to old adult rats? For example, 12 months and 18 months? What about their expression in PD rats?

2. Will progesterone treatment in PD rats improve PD pathological characteristics and the behavioral deficits?

3. Does PGRMC1 play a key role in progesterone treatment? What about other two receptors?  This should be addressed and discussed.

Author Response

Dear Editor,

Dear Reviewers,

We would like to thank you once again for inviting us to the publish our recent data on “Progesterone: a neuroprotective steroid of the intestine” and the critical review of the manuscript. Please find enclosed a point-by-point response to your comments. The manuscript was also proofread by a native speaker.

We sincerely hope that the manuscript in its present form will be accepted.

Yours sincerely

Carsten Theiss

Reviewer 1

Comments and Suggestions for Authors

Although PGRMC1 blockage experiment was added, the overall study is still too simple and lacks enough mechanistic explorations, especially in in vivo PD models. To strengthen the significance and novelty, further questions should be addressed, and additional experiments should be supplemented:

Thank you for the critical review of our paper. We understand your comments, but cannot fully agree with some points, which we would like to explain point by point. We do not believe that our study lacks novelty as, to our knowledge, we are the first research group to show receptor distribution in the ENS. While there are other publications on progesterone in the ENS (a Pubmed search yields 7 hits for the keywords “progesterone” plus “ENS”), we are aware of only one published study in which progesterone showed direct neuroprotective effects in the ENS. Nevertheless, this study does not address/discuss the corresponding receptors at all. Therefore, the analysis of progesterone receptor expression is the only logical first step and forms the fundamental basis for all further investigations.

  1. What is the expression of those progesterone receptors in ENS neurons of middle age to old adult rats? For example, 12 months and 18 months? What about their expression in PD rats?

Of course, we understand that this is an interesting question. However, we do not see this as being within the scope of this work. It would go beyond the scope of the current project and is also not feasible in 10 days. In addition, such a study would require animal testing applications, which are not currently available for this purpose. The formal and experimental procedure would delay the publication of the present data by up to two years. However, we would like to make these new and important results available to the scientific community as soon as possible, so that research on progesterone and the ENS as a whole can be advanced. We would like to reiterate that our data were obtained in an established in vitro PD model. Such published results justify the application and performance of animal experiments in the first place.

  1. Will progesterone treatment in PD rats improve PD pathological characteristics and the behavioral deficits?

Probably yes. At least there are promising results regarding progesterone therapy after traumatic brain injury in phase II and III clinical studies. As suggested by Reviewer 3, we have now added these data to the discussion, as they further support our thesis.

  1. Does PGRMC1 play a key role in progesterone treatment? What about other two receptors?  This should be addressed and discussed.

Our data strongly suggest that this is the case. Not only was the mRNA expression of PGRMC1 highest (see figure 2), but the newly added experiment allowed us to show that the neuroprotective effects of progesterone also disappear when this receptor is blocked. We have also already discussed this in detail in the following lines of our manuskript:

  1. 245-250: „Interestingly, PR-A/B showed the lowest mRNA expression levels. In addition, we detected higher relative expression levels of mPRa mRNA compared with mPRb. PGRMC1 showed the highest expression at the mRNA level across all ages tested. At the protein level, we detected colocalization of these progesterone receptors in myenteric ganglia at each developmental stage using confocal laser scanning microscopy.“
  2. 263-267: „However, just like in the ENS, the mRNA levels of both membrane receptors are lower than the expression levels of PGRMC1 mRNA in the CNS [32]. This comparison of relative receptor expression at the mRNA level might lead to the assumption that PGRMC1 is the predominant receptor mediating the rapid neuroendocrine effects in the ENS, as it is also discussed to be in most cerebral regions [32] and in the spinal cord [33].“
  3. 292-295: „Since the additional administration of the PCRMC1 antagonist AG205 abolished the neuroprotective effect in the current study, this suggests that the neuroprotective effect of progesterone is not genomic in nature and can be attributed at least in large part to the interaction of progesterone with PGRMC1.“
  4. 303-305: „However, while the exact mechanism remains to be further explored, there is strong evidence for an interaction of PAIRBP1 and PGRMC1 in mediating the neuroprotective effects of progesterone [44].“
  5. 316-320: „Further non-genomic mediated neuroprotective effects of PR-A/B could be explained due to the mitogen-activated protein kinase (MAPK) pathway previously demonstrated in the rat brain [51]. However, the observed attenuation of the neuroprotective effect of pro-gesterone after administration of the PGRMC-1 antagonist AG205 rather suggests a strong involvement of PGRMC1.“
  6. 329-331: „The decrease in cell death after progesterone treatment, may therefore also be explained by anti-oxidative effects of progesterone probably mediated by PGRMC1 [55].“
  7. 347-351: „The exact roles of each receptor in neuroprotective effects in this model appear to be very complex and cannot be fully elucidated with the current data. Nevertheless, our data strongly suggest that PGRMC1 plays a crucial role as a mediator of the neuroprotective effects of progesterone in the ENS. Further studies are required to improve the under-standing of progesterone-mediated control circuits.“

Reviewer 2 Report (Previous Reviewer 2)

I have reviewed the changes and comments of authors and I do not have any further comments. The authors have improved the manuscript and added experimental sections that improved the quality of the manuscript. I do agree with publishing in the current form.

Author Response

Dear Editor,

Dear Reviewers,

We would like to thank you once again for inviting us to the publish our recent data on “Progesterone: a neuroprotective steroid of the intestine” and the critical review of the manuscript. Please find enclosed a point-by-point response to your comments. The manuscript was also proofread by a native speaker.

We sincerely hope that the manuscript in its present form will be accepted.

Yours sincerely

Carsten Theiss

Reviewer 2

Comments and Suggestions for Authors

I have reviewed the changes and comments of authors and I do not have any further comments. The authors have improved the manuscript and added experimental sections that improved the quality of the manuscript. I do agree with publishing in the current form.

Thank you for the appreciation of our work and especially for the helpful comments and suggestions, which have greatly contributed to the improvement of our manuscript.

Reviewer 3 Report (Previous Reviewer 1)

I think that the authors did improve the quality of their paper by responding to my specific concerns. Hopefully, there will be further research done to study sex differences in progesterone efficacy in treating inflammatory and traumatic disorders of the nervous system. One minor point:  the authors stated that Phase II clinical trials with progesterone to treat TBI were inconclusive (see ref 58).  Actually, Phase II trials were successful and that's what led to the Phase III trials. This doesn't impact the present study, but the authors might still want to modify the statement in an otherwise excellent revision of their paper. Also, two, recent meta-analyses of clinical trials with neuroprotective agents, showed that of the 10 agents studied, progesterone + vitamin D and progesterone alone showed more efficacy in attenuating the effects of TBI than the other 9 neuroprotective agents used in clinical trials.  They might want to have a look at these recently published reports that would bolster the use of progesterone in enteric NS toxicity. The reports can be found on PubMed.

Author Response

Dear Editor,

Dear Reviewers,

We would like to thank you once again for inviting us to the publish our recent data on “Progesterone: a neuroprotective steroid of the intestine” and the critical review of the manuscript. Please find enclosed a point-by-point response to your comments. The manuscript was also proofread by a native speaker.

We sincerely hope that the manuscript in its present form will be accepted.

Yours sincerely

Carsten Theiss

Reviewer 3

Comments and Suggestions for Authors

I think that the authors did improve the quality of their paper by responding to my specific concerns. Hopefully, there will be further research done to study sex differences in progesterone efficacy in treating inflammatory and traumatic disorders of the nervous system.

One minor point:  the authors stated that Phase II clinical trials with progesterone to treat TBI were inconclusive (see ref 58).  Actually, Phase II trials were successful and that's what led to the Phase III trials. This doesn't impact the present study, but the authors might still want to modify the statement in an otherwise excellent revision of their paper. Also, two, recent meta-analyses of clinical trials with neuroprotective agents, showed that of the 10 agents studied, progesterone + vitamin D and progesterone alone showed more efficacy in attenuating the effects of TBI than the other 9 neuroprotective agents used in clinical trials.  They might want to have a look at these recently published reports that would bolster the use of progesterone in enteric NS toxicity. The reports can be found on PubMed.

We thank you for the encouraging comments from the first revision, which helped us a lot to improve this manuscript significantly. We were also happy to follow the specific recommendations from the last review. In addition, we have included the current results of the clinical studies on progesterone and traumatic brain injury in the discussion to further strengthen the manuscript.

see l. 351-362 – highlighted in red: “Still, a similar neuroprotective potential of progesterone in the CNS was postulated based on experimental animal models [58]. These promising results could also be reproduced in two phase II clinical studies, which examined progesterone administration after traumatic brain injury [59,60] but could not by confirmed in phase III clinical studies of traumatic brain injury [61,62]. What seems to be an inconsistency may in fact be attributed to various factors. On the one hand there are influencing factors regarding the injury, like its heterogeneity as well as possible comorbidities. On the other hand, factors regarding progesterone such as dosing, administration, and length of treatment, are found to be critical [reviewed in [63]]. Taken together, a comprehensive understanding of the regulation and mode of action of the different progesterone receptors in the ENS is essential, first in the ENS cell culture model, then in the animal model and finally in humans.”

Round 2

Reviewer 1 Report (Previous Reviewer 3)

/

This manuscript is a resubmission of an earlier submission. The following is a list of the peer review reports and author responses from that submission.

Round 1

Reviewer 1 Report

This is a well-written, novel and informative research paper showing that progesterone can protect enteric neurons exposed to the herbicide, rotenone.   The authors looked at a number of   potential, progesterone receptor activation mechanisms and also described the loss of ENS neurons caused by exposure to rotenone in neonatal rat tissue.  For this reviewer, there were no specific issues with the techniques employed or the interpretation of their results.  As the authors note, they did not examine potential for sex differences in response to treatment nor did they examine, in this study what might be the case in older, or even aged, animals exposed to rotentone. Further research is clearly needed to examine the morphological and functional effects of  rotenone toxicity in a live animal model and whether it can be attenuated by progesterone treatments or its metabolites.  This reviewer considers this work as a prelude to further research in this area, but it is an important first step that could open a new field of investigation and lead to better treatments when subjects are exposed to environmental neurotoxic agents that are used commercially. 

A few points need to be addressed.

  • The authors should provide the specific source for the progesterone used as treatment. Progesterone is not very soluable in aqueous solution so they might wish to comment on this.
  • It would be helpful to provide more details on the type of AoV they employed for their statistical analyses and what specific tests they used for individual comparisons between experimental groups. This should first be reported in Methods and not just in Fig. 5.

Reviewer 2 Report

The authors of the manuscript “Progesterone: A Neuroprotective Steroid of the Intestine” are describing the new neuroprotective potential of progesterone in the enteric nervous system.

The neuroprotective potential of progesterone is well described in the literature mainly targeting the central or peripheral nervous system. Yet the description of the expression of progesterone receptors in the ENS is valuable information. Moreover, the increased expression of PGRMC1 was described, suggesting the possible non-genomic action of progesterone. Accordingly, the progesterone was studied as a potential neuroprotective compound with the proposed indication of Parkinson´s disease. The viability of neurons after treatment with rotenone was shown.

Such an idea is of potential interest; however, the manuscript demonstrates several major limitations. The concept of this project should be reorganized and the manuscript should be re-written.

First, progesterone is a hormone by its nature. Then, a selection of this molecule only for testing is a major limitation as it is not clear what was the major target (ligand) of neuroprotection. As an expert on neurosteroids, their synthesis and their biological effect I can assure the authors that the proposed neuroprotection via progesterone receptors is very vague.
Such effect might be achievable also by another neurosteroid like estradiol, etc. Then, the expression of other nuclear receptors could be described.

Consequently, the concept of the manuscript would be the general neuroprotective potential of neurosteroids via ENS. Alternatively, such an effect may/may not be achievable only by neurosteroids and not neuroactive steroids (which I guess is not the case). Such information would be of great interest as once we know the major ligands for many neurosteroids/neuroactive steroids, we may speculate on the mechanism of action. Progesterone is a unique molecule that has been described as a target for so many ligands. The advantage of this molecule is the likelihood of success in the treatment. On the other hand, the proposed mechanism is missing as progesterone displays affinity toward so many neural targets. Then, a proposed relation towards neuroprotection of the intestine towards the expression of PGRMC1 and viability assay is simply not sufficient.
Second, I do miss the general concept of selection of rotenone. Indeed, other toxins are available and one could test everything that has in the lab. However, the authors of the manuscript are describing progesterone and the expression of progesterone receptors targeting the indication that is well known to be related to dopaminergic neurons. Indeed, not much is known about the neuroprotective effect of neurosteroids/neuroactive steroids on the dopaminergic system. Consequently, major links between the expression of dopaminergic neurons in ENS, the relation of PROG and the dopaminergic system, etc. are missing from your story.
As regards the claim of neuroprotection – the viability may serve as a hint for the neuroprotection. One might expect more scientific data, e.g. effect of ROS, caspases, etc.

Finally, only a concentration of 10 nM of progesterone was tested. This indeed suggests non-genomic action; however, it is known that once the concentration is increased, the non-genomic targets(ligands) might be saturated and the desired biological effect diminished.

Taken together, the described data are interesting, but after reading the manuscript, more questions are proposed than answered. Therefore, I do propose to reject this manuscript, collect more scientific data and rewrite the manuscript.

Reviewer 3 Report

The neuroprotective effects of progesterone in enteric nervous system have been well demonstrated in recent years. This is a simple in vitro study without any novelty at all, only a simple repeat of previous research with different pharmacological PD model. This paper lacks convincing in vivo study and necessary mechanistic investigation. Also, all the immunofluorescent images lack quantitative analysis. The overall quality of this paper needs to be further strengthened, especially addition of  mechanism study.